

# An intelligent zero trust secure framework for software defined networking

Xian Guo*, Hongbo Xian*, Tao Feng, Yongbo Jiang, Di Zhang and Junli Fang

School of Computer and Communication, Lanzhou University of Technology, Lanzhou, China
* These authors contributed equally to this work.

## ABSTRACT

Software-defined networking (SDN) faces many of the same security threats as traditional networks. The separation of the SDN control plane and data plane makes the controller more vulnerable to cyber attacks. The conventional "perimeter defense" network security model cannot prevent lateral movement attacks caused by malicious insider users or hardware and software vulnerabilities. The "zero trust architecture" has become a new security network model to protect enterprise network security. In this article, we propose an intelligent zero-trust security framework IZTSDN for the software-defined networking by integrating deep learning and zero-trust architecture, which adopts zero-trust architecture to protect every resource and network connection in the network. IZTSDN uses a traffic anomaly detection mode CALSeq2Seql based on a deep learning algorithm to analyze users' network behavior in real-time and achieve continuous tracking and analysis of users, restrict malicious users from accessing network resources, and realize the dynamic authorization process. Finally, the Mininet simulation platform is extended to build the simulation platform MiniIZTA supporting zero-trust architecture and the proposed security framework IZTSDN is experimentally analyzed. The experimental results show that the IZTSDN security framework can provide about 80.5% of throughput when the network is attacked. The accuracy of abnormal traffic detection reaches 99.56% on the SDN dataset, which verifies that the reliability and availability of the IZTSDN security framework are verified.

## INTRODUCTION

Since the article contains many abbreviations, the authors provide a table to sum up all of them in advance, the abbreviations are given in Table 1. The traditional TCP/IP network architecture is unsuitable for today's fast-growing network environment due to its complex configuration changes, inflexible resource scheduling, and high maintenance costs. In 2009, Professor Nick McKeown's team at Stanford University formally introduced the concept of SDN (*McKeown, 2009*). Nowadays, the application of SDN has been widely recognized as a new architecture, which is a widely adopted network architecture in IoT, Telematics, WAN, and 5G networks. SDN separates the control plane from the data plane, where the control plane is only responsible for decision forwarding, the data plane is only

Corresponding author
Xian Guo, iamxg@163.com

**Table 1 Abbreviations table.**

| Abbreviation | Full name |
| --- | --- |
| SDN | Software-defined networking |
| SDP | Software defined perimeter |
| ZTA | Zero trust architecture |
| UBTA | User behave trust authentication |
| IZTSDN | Intelligent zero trust software defined networking |
| CALSeq2Seq | CNN self-attention Lstm-Seq2Seq |
| MiniIZTA | Mini intelligent zero trust |

responsible for routing and forwarding, and the application plane provides open and programmable services for users. The central controller of SDN provides a flexible configuration of network devices, which improves network service flexibility and management efficiency, greatly reduces operation and maintenance costs, and enables network performance scaling (*Nisar et al., 2020*; *Keshari, Kansal & Kumar, 2021*; *Yurekten & Demirci, 2021*).

However, SDN faces the same threats of many network attacks as traditional networks (*Chica, Imbachi & Vega, 2020*). The controllers in SDN are not designed with the security of the controllers themselves as a research component, which makes SDN controllers vulnerable to similar distributed denial-of-service attacks and port scanning attacks (*Santos et al., 2020*; *Singh & Behal, 2020*). Remote work and multiparty collaboration have become a norm, making the complexity of access requirements higher and leading to the risk of expanding the attack surface of internal resources (*Weng et al., 2018*; *Fang et al., 2019*). There is also a lack of strict traffic detection mechanisms in the communication between switches and controllers and service resources, and malicious network attacks may cause flow table overflows and congestion (*Zhang, Tang & Barolli, 2019*).

The traditional "perimeter defense model" can not prevent internal lateral movement attacks caused by malicious internal users or software vulnerabilities, and only determines whether users' requests are trusted by dividing the internal and external security zones. The coarse-grained access control makes the inner security zone over-trusted (*Gold, 2014*). In 2010, John, a principal analyst at Forrester, formally introduced the term zero-trust and clarified the idea of "never trust, always verify", and in 2020, the National Institute of Standards and Technology published the Zero Trust Architecture (ZTA) standard (*Kindervag, 2010*; *Kerman et al., 2020*). The main feature of zero trust architecture is "authentication before connection", which shifts from static network defense based on network boundary location to protecting each network resource and each network connection, and each access request will be continuously authenticated and authorized to achieve fine-grained access control. Software defined perimeter (SDP) (*Moubayed, Refaey & Shami, 2019*) is a classic solution proposed by the Cloud Security Alliance to achieve a ZTA. In a SDN security scheme based on zero-trust architecture, how to achieve dynamic and continuous authentication and authorization is the focus of the research.

Researchers have proposed a large number of machine learning-based intrusion detection schemes (*Garg et al., 2019*; *Ravi, Chaganti & Alazab, 2022*) for software-defined networking, however, most of the existing literatures on software-defined network security research considers the detection of anomalous traffic in SDN by improving the accuracy of machine learning algorithms, but due to the difficulty of data acquisition, some researchers use data sets from traditional networks to train the algorithm models, which are hardly applicable to the model is hardly applicable to SDN networks. Some researchers only propose a security mechanism to secure the central controller but do not consider the security status of the whole network, which has certain limitations.

To address the above problems, this article combines deep learning and zero-trust architecture and proposes an Intelligent Zero Trust security framework for SDN (IZTSDN), which consists of five modules: data collection module, trust assessment engine module, intelligent behavior analysis engine module, intelligent controller module, and communication execution module.

The data collection module collects the historical behavioral data of users to provide data support for authentication and authorization decisions when the intelligent controller processes user access requests to network resources, so as to better evaluate the legitimacy of user-initiated requests. The trust assessment module uses the trust authentication algorithm to calculate the predicted user's "trust value" based on the data provided by the data collection module, which is used as an important basis for the intelligent controller module to achieve dynamic authentication and authorization of the user's application for accessing network resources. The intelligent behavior analysis engine module detects the user's network behavior in real-time, and uses a deep learning model deployed on the SDN controller to analyze and evaluate the behavior and security status of network-connected users in real time; The intelligent controller module grants users access to network resources based on the user trust value calculated by the trust assessment module, opens the list of network resources accessible to the requesting user, and notifies the gateway of the user's legitimate connection request information to achieve dynamic authentication of legitimate users and fine-grained access control, at the same time, the intelligent controller module detects and judges the user's network behavior based on the intelligent behavior analysis module. In the communication execution module, users obtain access rights to network resources and a list of accessible resources, then initiate a request to the gateway to connect to the network resources, and the gateway establishes a two-way authentication communication channel between the user and the network resources in cooperation with the user based on the legitimate user connection request information from the intelligent controller module to achieve secure access to the network resources by the user.

The main contributions of this article are as follows:

(1) Integrating Deep learning and zero-trust architecture to propose IZTSDN, an intelligent zero-trust security framework for software-defined networking, which aims to be able to prevent internal and external network attacks and protect every network resource and network connection;

(2) The IZTSDN framework introduces a traffic anomaly detection model CALSeq2Seq for the detection of anomalous traffic in SDN. The model is evaluated by using the publicly

available SDN dataset from Bennett University Research Institute, and the experimental results show that it outperforms other algorithmic models in several evaluation metrics, including accuracy, false alarm rate, precision, recall, and F1-measure, and also compares the models proposed by other researchers, and the results all show that the CALSeq2Seq model has a high accuracy rate and a low false alarm rate.

(3) IZTSDN framework introduces a trust algorithm to evaluate the user's trust value based on the user's historical behavior data, which is used as one of the important bases for authentication and authorization to access network resources and deep learning algorithm is used to analyze user's network behavior in real-time to realize continuous tracking analysis of users, restrict malicious users from accessing network resources, and realize dynamic authentication and authorization;

(4) This article extends the Mininet simulation tool to build a simulation platform MiniIZTA sup-porting zero-trust architecture, which can be used for network protocol analysis and verification under zero-trust architecture.

The rest of the article is organized as follows: The background significance of the research topic is introduced in the "Introduction". The current state of research on cyber attacks in SDNs is presented in "Related Work". The proposed security framework IZTSDN is introduced in "Intelligent zero-trust security framework for software-defined networking", and the functions of the five core modules are also described in detail. In "Simulation Experiments", we describe how to integrate zero-trust components to build the simulation platform MiniIZTA, and evaluate the performance of IZTSDN in terms of two metrics: network throughput and traffic anomaly detection accuracy. Finally, the main research contents of this article are summarized in "Conclusion".

## RELATED WORK

In recent years, it has been a hot topic of research on how to solve network attacks in SDN, and many researchers have conducted in-depth studies on the existence of simple authentication methods in SDN that lead to internal malicious users launching network attacks and frequent DDoS and port scanning attacks on SDN controllers. Many intelligent architectures and anomalous traffic detection frameworks have been proposed in the existing literature to address the threats of security attacks in SDN.

NetworkAI (*Yao et al., 2018*) is a network intelligence architecture in SDN that can learn control policies autonomously. In the control plane, the logical and centralized control of the forwarding plane, which is mainly responsible for giving an analysis of the network traffic and state, allows the generated control policies to be quickly deployed to the network to achieve intelligence. A security framework for protecting SDN, the SDN-ecosystem (*Carvalho et al., 2018*), which uses a traffic collection module to collect data, followed by a detection module to determine whether the traffic is normal or not, and subsequently mitigates abnormal traffic mainly by directly discarding or change the flow table entry, and finally the reporting module is mainly responsible for displaying the attack report, the disadvantage is that the algorithm complexity is large and easy to cause CPU overload. A cross-plane DDoS attack defense framework OverWatch in SDN (*Han et al., 2018*), which can collaborate between the data plane and control plane to defend against

attacks. In the article, the authors propose a DDoS attack detection mechanism that consists of two main algorithms, which are a fine-grained machine learning attack classification algorithm on the control plane and a coarse-grained traffic monitoring algorithm on the data plane. A defense mechanism is also proposed, which is deployed on controllers and switches, thus enabling rapid defense. Regarding the issue of software cognition and the application of fuzzy intelligent learning, researchers have also proposed corresponding solutions to address the existing security issues (*Haleem, Farooqui & Faisal, 2021a*, *2021b*; *Zhang et al., 2022*).

In addition to the aforementioned researchers using various security mechanisms to propose a security framework, subsequent researchers have successively introduced deep learning methods to detect malicious attacks appearing in the network. A fusion learning security scheme between the data plane and control plane in SDN environments (*Selvi & Thamilselvan, 2022*). Prediction is provided through model parameter exchange in the SDN client model and data distribution in a single communication. The SDN controller uses a deep neural network GRU model. GRU captures only temporal dependencies. To handle the dynamics of network traffic and efficient capture of traffic patterns, a diffusion convolution operation is also embedded to capture the spatial and temporal dependencies of features in the encoder-decoder architecture, enhancing the network traffic prediction. An SDN-enabled solution driven (*Javeed, Gao & Khan, 2021*) by deep learning algorithms to detect network threats and attacks using algorithms consisting of CuDNNLSTM and CuDNNGRU, with hybrid intrusion detection models placed on the programmable control plane of the SDN. The proposed algorithm is trained in experiments using the publicly available dataset CICDDoS 2019, and the results show that the algorithm has a minimal false alarm rate and high accuracy identification rate. A highly scalable SDN-enabled malware detection framework (*Khan & Akhunzada, 2021*) that utilizes a hybrid DL architecture (CNN-LSTM) for medical IoT. A comprehensive evaluation was conducted in experiments using current state-of-the-art publicly available IoT datasets and evaluated using standard performance evaluation metrics. The experimental results show that the proposed framework performs well in terms of detection accuracy and computational complexity. A modular, flexible, and scalable SDN security framework (*Yungaicela-Naula et al., 2022*) that integrates a deep learning algorithm-based intrusion detection system and a deep reinforcement learning-based intrusion prevention system to deal with slow DDoS attacks. The proposed lightweight intrusion prevention system based on deep reinforcement learning can provide fast mitigation response. After extensive experiments, the proposed SDN security framework achieves an average detection rate of 98% and timely mitigation of slow DDoS.

With the strong promotion of the zero-trust security concept worldwide in recent years, there are more and more scholars engaged in zero-trust-related research and related results at home and abroad. There are also some studies on using zero-trust architectures to address SDN security, as both are implemented in the form of software-defined programmability and have great similarities. An integrated SDP-SDN architecture (*Sallam, Refaey & Shami, 2019*) that provides a better security platform. Experiments show that it can block PS and DDoS attacks while still maintaining 75% of network throughput

without disruption, but lacks a dynamic and continuous user authentication mechanism. A framework (*Moubayed, Refaey & Shami, 2019*) for a zero-trust architecture using a client gateway architecture and evaluate its performance using a virtual network testbed for an internal enterprise scenario as a use case. The performance evaluation results show that despite the long initial connection establishment time, the zero-trust secure network is highly resistant to denial-of-service attacks and port scanning attacks.

Domestic and international researchers have found that in such a complex and dynamic network environment, traditional network security frameworks have obvious weaknesses and are difficult to provide adequate security. The proposal of Zero Trust breaks the previous traditional authentication approach, which is a security concept for meeting the security requirements of networks with untrusted infrastructures. The concept of intelligent zero-trust architecture (*Ramezanpour & Jagannath, 2022*) is based on which artificial intelligence algorithms can be applied to provide information security in untrusted networks. Provide information security in untrusted networks. The implementation supervises the state of network resources, evaluates each user's access request, and uses dynamic trust algorithms to determine access authorization.

Existing studies have shown that this combined model approach can significantly improve the accuracy of deep learning algorithms. A scheme (*Ujjan et al., 2020*) based on sFlow and adaptive polling sampling combined with a deep learning model to reduce various DDoS attacks within IoT networks. A TCP-SYN and ICMP flood attack mitigation approach using machine learning methods (*Tuan et al., 2020*) in SDN-based Internet Service Provider (ISP) networks. Extensive experimental results show that the algorithm can effectively mitigate more than 98.0% of the attacks without affecting benign traffic. A new method (*Banitalebi Dehkordi, Soltanaghaei & Boroujeni, 2021*) combining statistical and machine learning techniques that can be used to detect both high-volume and low-volume DDoS attacks. The importance of the method in terms of accuracy is that it outperforms similar methods. A proposed DDoS attack detection method (*Liu et al., 2019*) combining generalized entropy with PSO-BP neural network for DDoS attacks in SDN. Based on this A deep learning approach for DeepIDS, an intrusion detection system for SDN networks (*Tang et al., 2020*). By using the NSL-KDD dataset for training and testing, a fully connected deep neural network DNN and a gated recurrent neural network GRU-RNN were used to achieve the detection target, achieving 80.7% and 90% accuracy, respectively. In addition, the system performance was evaluated in terms of throughput, latency, and resource utilization. The test results show that DeepIDS does not affect the performance of the OpenFlow controller and provides significant performance improvements in sequence learning, but there is still room for further improvement in detection rate and accuracy. A detection and defense system (*Novaes et al., 2021*) based on SDN adversarial training. The system uses a generative adversarial network GAN framework for anomaly detection of DDoS attacks on SDN in real time. A detection method (*Cao et al., 2021*) based on a spatiotemporal graph convolutional network (ST-GCN). This method detects the network state of a switch and feeds it into a spatiotemporal graph convolutional network detection model to finally find the switch through which the DDoS attack flow passes.

This article focuses on the analysis and research of network attacks faced in SDN, proposes the corresponding deep learning detection algorithm CALSeq2Seq and trust authentication algorithm UBTA for the identification of abnormal traffic in SDN and the problems of traditional authentication methods, and proposes an intelligent zero-trust framework IZTSDN in order to achieve all-round protection of SDN, and introduces the previously proposed two algorithms in its intelligent engine plane to assist the central controller to issue control policies to achieve intelligence. The proposed two algorithms are introduced in the intelligent engine plane to assist the central controller to issue control policies to achieve intelligence, and the entire framework is completed by the zero-trust software-defined boundary core components. In addition, the zero-trust core components are integrated into Mininet to build the simulation platform MiniIZTA in order to realize the intelligent zero-trust security framework IZTSDN proposed in Chapter 4, and finally, the security and reliability of the framework are verified through simulation experiments.

# INTELLIGENT ZERO TRUST SECURE FRAMEWORK FOR SOFTWARE DEFINED NETWORKING

This section details IZTSDN, a software-defined network security framework based on the intelligent zero-trust architecture proposed in this article, which mainly applies the intelligent zero-trust architecture in SDN to solve the existing security problems. The IZTSDN framework is completely based on the core idea of zero-trust architecture and is based on deep learning methods to solve known and unknown security problems in SDN networks and prevent malicious users from launching lateral movement attacks by controlling a node. The general architecture of IZTSDN is given in Fig. 1.

The IZTSDN framework consists of five modules whose respective functions are described as follows:

(1) Data collection module: This module collects historical user behavior data from multiple dimensions (log system, system configuration information, access request information, *etc.*) to provide data support for authentication and authorization decisions when the intelligent controller processes user access requests to network resources in order to better evaluate the legitimacy of user-initiated requests.

(2) Trust assessment engine module: This module adopts the dynamic trust authentication algorithm (User Behave Trust Authentication, UBTA) to calculate the predicted user's "trust value" based on the data provided by the data collection module, which is used as the intelligent controller module to realize the dynamic authentication and authorization of the user's application. The value is used as an important basis for the intelligent controller module to achieve dynamic authentication and authorization for users to access network resources.

(3) User behavior analysis engine module: This module uses the traffic anomaly detection model CALSeq2Seq to detect attack traffic in the SDN network. The model first uses one-dimensional convolutional neural networks (1DCNN) to extract network features from the collected data, then introduces a self-attentive mechanism to achieve the
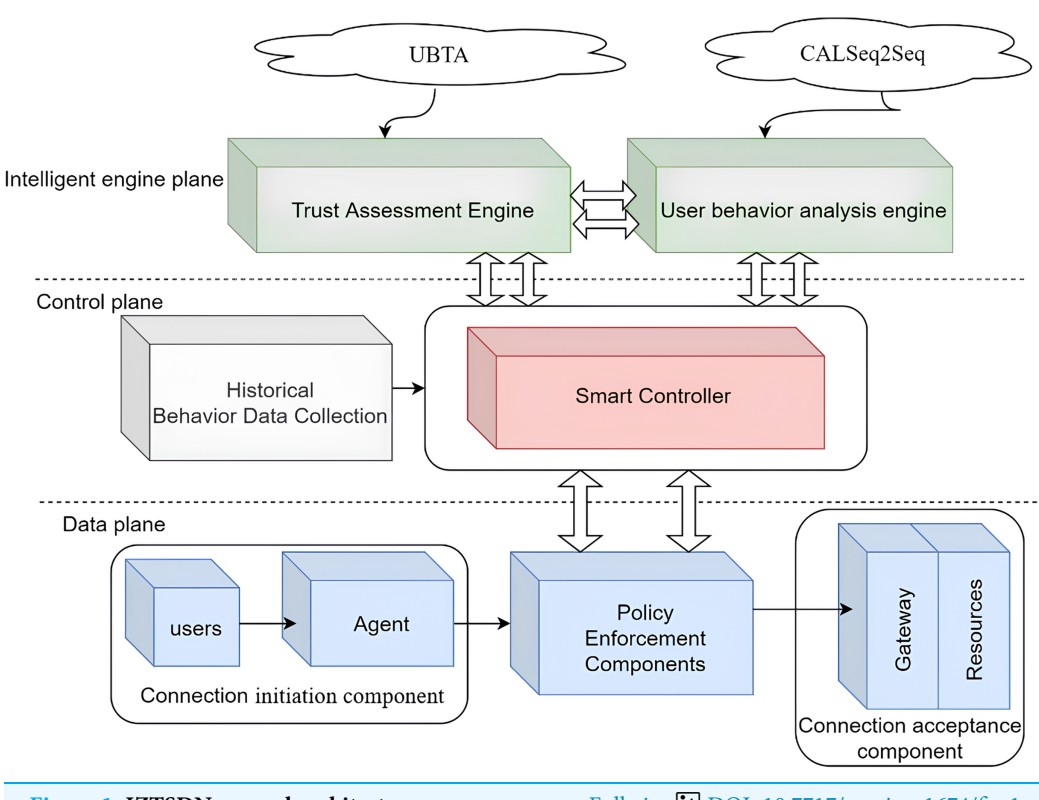

**Figure 1 IZTSDN general architecture.**

fusion of key features, and uses the fused high-level features as the input parameters of the LSTM-Seq2Seq structured network. The long short-term memory network (LSTM) is used as the codec of Seq2Seq to learn the dependencies between the features, and finally, the prediction results are output through the fully connected layer. The model analyzes and evaluates the behavior and security status of the network-connected users in real-time.

(4) Intelligent controller module: This module grants users access to network resources based on the user trust value calculated by the trust assessment module and the corresponding authentication and access control policies, and opens the list of access network resources to the requesting user, notifies the gateway of the user's legitimate connection request information, and realizes dynamic authentication as well as fine-grained access control for legitimate users. At the same time, this module, based on the detection and judgment of user network behavior by the behavior analysis module, tracks and blocks illegal operations initiated by malicious internal users in real-time to achieve the purpose of protecting each network connection.

(5) Communication execution module: After users obtain access rights to network resources and a list of accessible resources, they initiate a request to the gateway to connect to the network resources. Based on the legitimate user connection request information from the intelligent controller module, the gateway cooperates with the user to establish a two-way authentication communication channel between the user and the network resources to achieve secure user access to the network resources.

## Data collection module

To ensure the security and reliability of SDN systems, historical behavioral data needs to be meticulously collected and analyzed. This data covers a variety of information, including user logs, system status information, activity logs, network traffic, resource access operations, and other events, as well as enterprise user information and identity records. By collecting this information and unifying it for processing, dynamic updates of end-user trust values can be achieved, and assist the IZTSDN security framework in discovering potential security threats. To acquire network data from multiple dimensions, the data collected is divided into two parts: SDN flow table data and service flow sequences. Network state information inside the SDN, including flow table information and port statistics, *etc.*, is obtained in real-time through the OpenFlow protocol, and matching protocol fields in the flow table (such as IP address and port number), as well as flow table-specific fields (such as counters), are extracted. Also, collecting service traffic data, including flow table message data and flow table interval data, enables better detection of attack traffic.

When collecting traffic data, the sampling method used needs to maximize the detection of the way users request services. This is because the flow table data in normal actual user access requests are basically composed of consecutive transactions, each consisting of multiple consecutive requests. In subsequent traffic inspection, a certain slice can be selected to extract the basic characteristics of network traffic and distinguish it from the traffic of simulated attacks. This allows better detection of anomalous traffic and reduces potential security risks.

## Trust assessment engine module

The trust authentication algorithm UBTA based on user behavior is proposed in the trust evaluation engine module. UBTA extracts valid features from user behavior (data collection), analyzes and predicts its real-time trust value, and grants corresponding privileges according to its given operation threshold to provide more fine-grained access control to the accessing users and achieve dynamic continuous authentication to stop unauthorized attacks and intranet attacks.

By inputting the historical behavioral data of access users to correspond to the corresponding evaluation factors and weights from multiple dimensions, the trust value of current access users is calculated by their continuous behavioral data, and the trust value of access users is dynamically updated according to the introduced time decay factor. The common behaviors generated by users are given in Table 2.

### The input of historical data

The user's historical data is mainly determined by the trust evaluation factor and weight in the trust algorithm. the trust evaluation factor mainly corresponds to the trustworthiness of the current user, 0 represents no trustworthiness, 0.5 represents temporary trustworthiness, and 1 represents complete trustworthiness. This factor is determined

**Table 2 User behavior characteristics.**

| Behavior type | Specific behavior |
|---|---|
| Login behavior | Login method, time, duration of access, device, IP address |
| Network behavior | Upload traffic, download traffic, TCP, UDP, and ICMP traffic |
| Operation behavior | Name of resources accessed, history of operations |

according to the degree of deviation of user behavioral data. T denotes the continuity of the data in the duration, and t denotes the continuous login time. As shown in Eq. (1).

$$T = \{t_1, t_2, t_3, \ldots\ldots, t_{n-1}\} \tag{1}$$

### Calculation of the current trust value of the accessed user

The current trust value of the accessed user is calculated from the mean and standard deviation of the continuous login time, and the current login deviation is normalized. This is shown by Eq. (2).

$$t' = \left| \frac{t - u}{\sigma} \right| \tag{2}$$

If the behavioral data deviation is small, the trust factor is set to 1. If the behavioral data deviation is within a certain range, the trust factor is set to 0.5; if the behavioral data deviation is too large, the trust factor is set to 0.

The user's current trust value is calculated by assigning an initialized weight proportion to each feature dimension and subsequently multiplying it with the trust evaluation factors of each user dimension and then adding them together. This is shown by Eq. (3).

$$Trust = \sum_{i=1}^{n} S_i w_i \tag{3}$$

$n$ is the dimension of behavior characteristics, $S_i$ is the trust evaluation factor of each dimension, $w_i$ is the weight contained in each dimension. This weight is mainly adjusted dynamically through its system state in order to better calculate the current trust value of users. Trust thresholds corresponding to operations are given in Table 3.

### Predict users' real-time trust values

The accessing user usually has multiple access behaviors, and the user's historical trust should introduce a time decay factor $\gamma$ to the current trust. The value of $\gamma$ between 0 and 1. The dynamic trust of the user in the current cycle $Pred_k$ is the sum of the trust of the previous cycle $Pred_{k-1}$ and the trust of the currently visiting user $Trust$.

$$Pred_k = \gamma Pred_{k-1} + (1 - \gamma) Trust \tag{4}$$

when the user's real-time trust value is greater than the trust threshold of an operation, the system will assign the user the corresponding permission. if the user's current behavior

**Table 3 Trust thresholds corresponding to operations.**

| Operation | Confidence threshold |
|---|---|
| Browse view | 0.5 |
| Download file | 0.7 |
| Upload a file | 0.8 |
| Modify file | 0.9 |
| Delete file | 0.95 |

trust value is 0.55, the user can only be authorized permission to browse. If the user's behavior trust value is 0.75, the user can not only browse but also have permission to download the resource, if the user's behavior trust value is 0.85, the user can add an additional permission to upload, if the user's behavior trust value is 0.9 or more, then the user will have all the operations on the resource. The behavioral trust of access users is continuously and dynamically calculated throughout the network system, while the trust threshold of each permission is dynamically adjusted according to the environment to achieve more fine-grained and intelligent access control. The authentication process is given in Fig. 2.

Trust assessment is to allow legitimate users to access the resources normally. First, let the user initiate a valid SPA data request packet, and use the UBTA algorithm to calculate the user's trust value According to the trust threshold to grant its access rights, the intelligent controller will send a message to the gateway, making the establishment of access rules flow table so that the gateway can allow the authenticated user to access the resources normally within the specified time period T. In this process, the gateway will continuously receive the intelligent controller to send In this process, the gateway will continue to receive messages from the intelligent controller for user authentication, in the communication process must establish a two-way encrypted channel communication MTLS connection to ensure data transmission security.

## User behavior analysis engine module

The highly centralized control plane of SDN is often subjected to various network attacks, which makes the SDN controller prone to paralysis or downtime, resulting in fundamental damage to the entire network environment. It is necessary to design an efficient and accurate traffic anomaly detection model for SDN.

After research, it is found that most of the current methods are relatively single deep learning models, which have some limitations in traffic feature extraction. For example, for some complex traffic characteristics, a single deep learning model may not be able to capture all of its information, resulting in incomplete feature expression. In addition, the real-time performance of a single model is weak, and the analysis of network traffic needs real-time processing. However, a single deep learning model may have problems such as long training time and slow prediction time, which will seriously affect the accuracy of model recognition and classification. In order to better detect the anomaly of attack traffic in SDN, a traffic anomaly detection model based on CNN self-attention LSTM-Seq2Seq

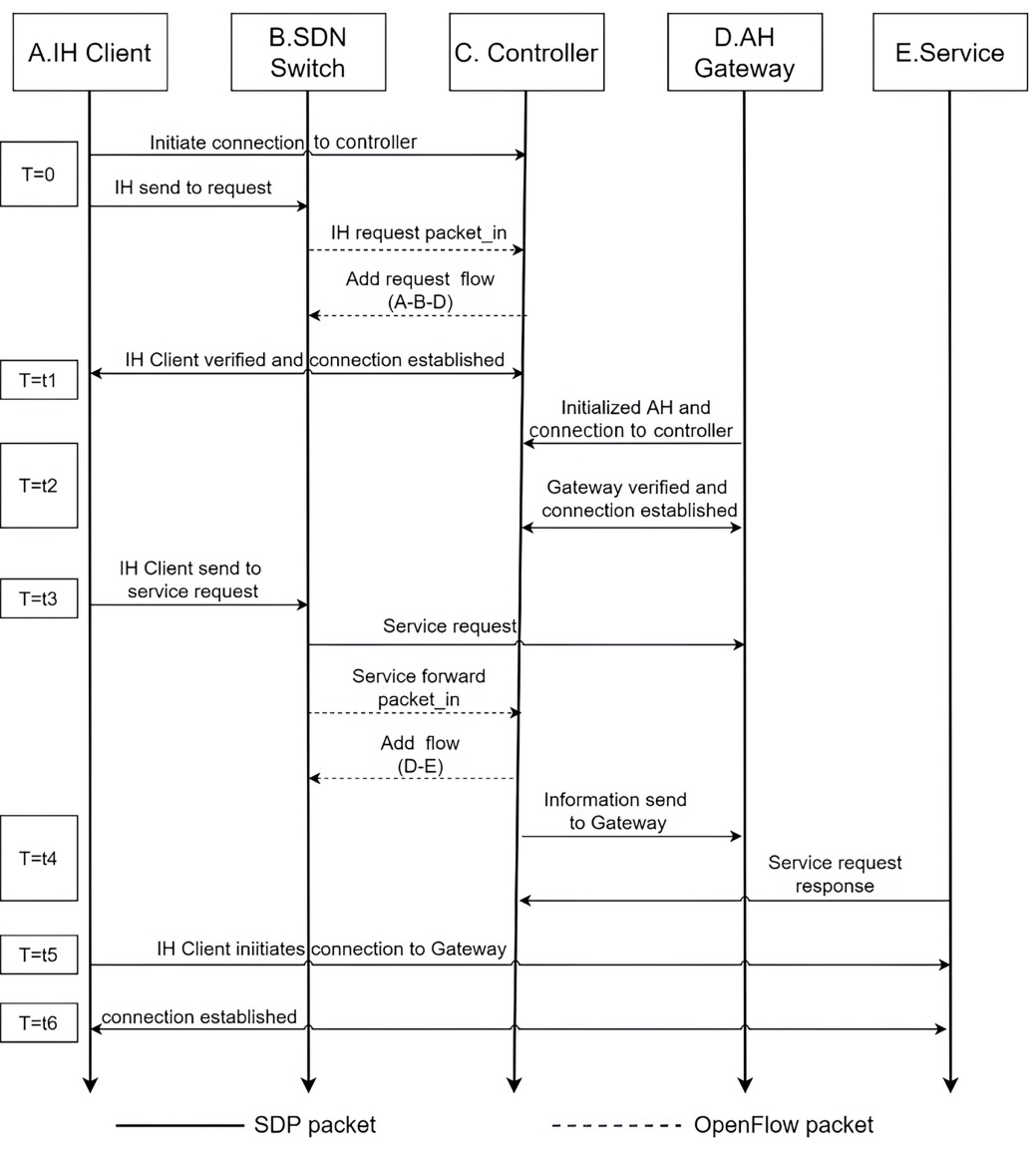

**Figure 2 Timing diagram of user access to resources.**

and CALSeq2Seq is proposed, and the data preprocessing module, feature extraction module, feature fusion module and detection and classification module are designed.

The CALSeq2Seq model algorithm uses 1D CNN to extract features from the original data in the feature extraction stage, flattens the extracted feature map into a long vector, introduces the self-attention mechanism to fuse the key features, and uses the high-level features after the feature fusion stage as the input of the LSTM-Seq2Seq structure encoder. These are then decoded to output for timing modeling. The CALSeq2Seq model is given in Fig. 3.

The main innovations of the model are as follows: the self-attention mechanism is introduced to extract key features, and the obtained advanced features are used as the input of the detection and classification module to improve the prediction and classification

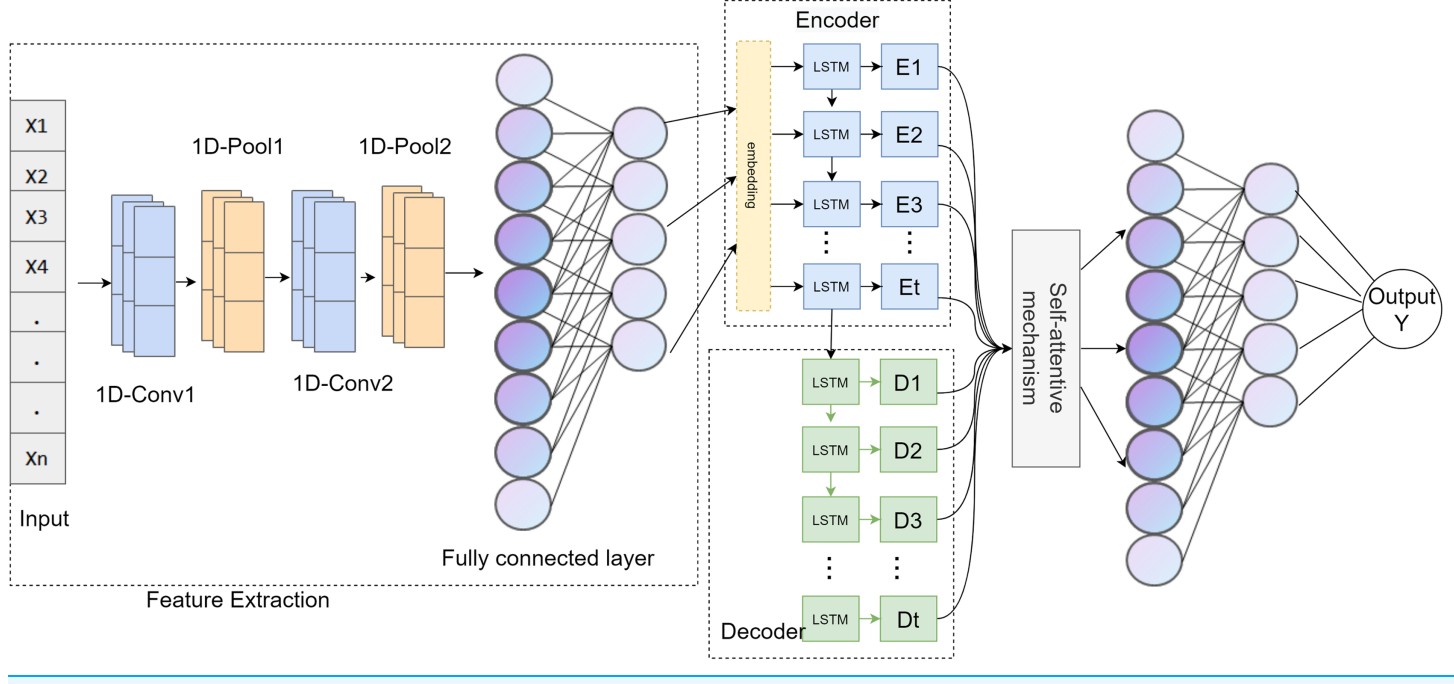

**Figure 3 CALSeq2Seq model.**

ability of the model for network abnormal traffic. In the detection and classification module, a Seq2Seq model based on LSTM as encoder-decoder is used. These LSTM units model the dependencies of existing sequence data, and the accuracy and detection speed are significantly improved.

The traffic anomaly detection model CALSeq2Seq is shown in Fig. 3, which is mainly composed of data preprocessing module, feature extraction module, feature fusion module and detection and classification module to complete the prediction output of network traffic. In the SDN flow table, according to its specific OpenFlow protocol format, multiple flow bytes are combined into a flow packet, and multiple flow packets are combined into a network flow. CALSeq2Seq, a deep learning algorithm traffic anomaly detection model, is used to detect whether the network flow is normal. When the device node in the data plane gives up the access request, the data acquisition module obtains the request traffic. After data preprocessing, the 1D CNN is used to extract the effective traffic features, and then the self-attention mechanism is introduced for feature fusion. The fused high-level features are used as the input of the encoder process to the LSTM-based Seq2seq to flatten the extracted feature map into a long vector for predictive modeling. Finally, the decoder output vector was mapped by the fully connected layer according to the task of the output layer, and the SoftMax activation function was used to map the fused high-level features to the classification result (normal/abnormal). As indicated by Algorithm 1.

Time consumption and complexity analysis:

We have utilized hardware acceleration in training the model to speed up the whole model training, probably each step can probably be controlled to a very small amount of

---

**Algorithm 1** CALSeq2Seq-based traffic anomaly detection model.

Input: training data set X input n samples, each sample is $x^i$, where $i \in (1, \ldots, n)$, initialize the LSTM unit

Output: Identification classification Y

Data pre-processing, abnormal data processing, eigenvalue coding, normalization

Begin

$\quad O_i = ReLU(W * f_i + b)$      // uses ReLU to obtain output features

$\quad G = QK^T\sqrt{d_k}$      // Introduction of self-attentive mechanism

$\quad W = soft\,max(G)$

$\quad Z = Attention(Q, K, V) = WV$

$\quad E_t = LSTM_{en}(x_t, E_{t-1})$      // Input Z to the LSTM cell to get the output category Y

$\quad D_t = LSTM_{dn}(x_t, D_{t-1})$

$\quad softmax(Z_j) = e^{z_j}/\sum_{K=1}^{K} e^{z_k}, j = 1, \ldots, k$

$\quad Y = argmax[softmax(z_j)]$      // Return the predicted result

End

Return Y

---

less than 10 s, although it took some time in the training phase, but achieved good results in the subsequent model prediction phase, according to Algorithm 1, this time complexity can be optimized to about O(n) at present.

### Data pre-processing module

The acquired data usually have abnormal data (missing values or duplicate values), so in order to better explore the relationships existing between the data, data pre-processing must be performed before using the data. The main steps include the following:

Abnormal data processing: If certain features are found to have missing values in the data, such features need to be removed to prevent errors in data analysis.

Feature value encoding: There are more discrete and disordered feature values in the data, and one-hot encoding is needed to help the algorithm model to better train as well as output results, which plays the role of expanding features.

Min-Max normalization: the linear transformation of the original data, so that the resultant values are mapped to between [0,1]. Min-Max normalization reduces the time required to converge to the local minima and improves the model's accuracy.

### Feature extraction module

After anomalous data processing, eigenvalue coding, normalization, and data reshaping in the data preprocessing module, the 1D traffic data of length $l$ is reshaped into a 2D matrix M with height $h$, width $w$, and a number of channels 1. Next, a 1D CNN is used in the feature extraction module to extract the effective features, and two convolutions and two maximum pooling operations are used to extract the features.

The first convolution operation uses 32 convolution kernels of size $3 \times 3$ to convolve the input tensor, and each convolution kernel generates one output channel. The output tensor is then shaped as $(h - 2, w - 2, 32)$, and the output tensor is pooled with a maximum of $2 \times 2$.

In the second convolution operation, 32 convolution kernels of size $3 \times 3$ are used to convolve the output tensor of the first pooling operation, and each convolution kernel generates one output channel. The output tensor is then max-pooled by $2 \times 2$.

After two convolutions and two pooling operations, the final 3D tensor is obtained, and this 3D tensor is represented as a matrix consisting of several 2D slices, where the Kth 2D slice corresponds to a set of feature vectors extracted from a region of the input, denoted as $x_k$, The Flatten layer is used to flatten the multidimensional input data into a one-dimensional vector in order to feed it into the neural network for processing, spreading the resulting matrix into a one-dimensional vector $f$ shown by Eq. (5).

$$f = [x_1, x_2, ..., x_n]^T \tag{5}$$

This one-dimensional vector $f$ is the input passed to the fully connected layer. After receiving this input, the fully connected layer multiplies it with the weight matrix $W$ and adds the bias vector $b$ to obtain the final output vector $O_i$ shown by Eq. (6).

$$O_i = ReLU(W * f_i + b) \tag{6}$$

### Feature fusion module

In order to further strengthen the long-term memory capability of LSTM and make it more perceptive to long-distance information, the self-attentive mechanism is introduced to have a stronger ability to capture long-term dependencies. The output features obtained from the feature extraction stage are used as the input of the self-attentive mechanism, and three transformation matrices $Q$ (Quer, query), $K$ (Key, key), and $V$ (Value, value) are obtained, and the attention score matrix G is obtained by the operation of scaling the dot product, where $d_k$ is the dimension of Key and $1/\sqrt{d_k}$ is the scaling factor to prevent the inner product value from being too large to affect the training of the neural network, as shown in Eq. (7).

$$G = \frac{QK^T}{\sqrt{d_k}} \tag{7}$$

The attentional weight matrix is shown by Eq. (8). The attention weight matrix and $V$ are multiplied to obtain a result matrix Z, which incorporates the self-attentive mechanism, shown by Eq. (9).

$$W = softmax(G) \tag{8}$$

$$Z = Attention(Q, K, V) = WV \tag{9}$$

### Detection and classification module

The LSTM-based Seq2Seq model plays a role in the detection classification module for the predictive classification of traffic generated by user behavior. The model uses an encoder-decoder framework to map the input sequence of values to the output sequence. The encoder converts the high-level features fused by the pre-processing module and the self-attentive mechanism as input sequences into fixed vectors, which are then decoded into output sequences. The encoder in the model consists of LSTM units that attempt to model the dependencies between past and present sequence data, encoding fixed lengths that are also decoded using a sequence of LSTM units.

The LSTM sequential model is easier to learn long-term dependence than simple recurrent architecture and solves the problem of gradient disappearance and gradient explosion during training. The LSTM sequential model contains three gates: input gate ($i_t$), output gate ($o_t$), and forgetting gate ($f_t$), and the weight is set by the neural network layer sigmoid to a value between 0 and 1, which describes how many input signals can be passed. means "no amount is allowed to pass", 1 means "all amount is allowed to pass". These three gates are used to control the information retention and transmission of the LSTM, which is eventually reflected in the cell state $C_t$ and the output signal $h_t$, These three gates are represented by Eqs. (10)–(12).

$$i_t = \sigma(w_i[h_{t-1}, x_t] + b_i) \tag{10}$$

$$o_t = \sigma(w_o[h_{t-1}, x_t] + b_o) \tag{11}$$

$$f_t = (w_f[h_{t-1}, x_t] + b_f) \tag{12}$$

$w_x, b_x, x \in \{i, o, f\}$, denote the weights and deviations of the three gates, $h_{t-1}$ denotes the output of the previous LSTM unit, $x_t$ denotes the current output. The memory gate consists of an input gate ($o_t$) with a tanh neural network layer and a per-bit multiplication operation, and Eq. (13) represents:

$$\tilde{C}_t = tanh(w_C[h_{t-1}, x_t] + b_C) \tag{13}$$

Here, the output $f_t$ of the forgetting gate is multiplied with the cell state $C_{t-1}$ at the previous moment to select forgetting and retain some information, and the output of the memory gate is summed with the information selected from the forgetting gate to obtain the new cell state $C_t$, Eq. (14) represents:

$$C_t = f_t * C_{t-1} + i_t * \tilde{C}_t \tag{14}$$

The output gate is a sigmoid neural network layer that outputs a value $o_t$, between 0 and 1 after the previous forgetting gate and memory gate processing, which can obtain the prediction results, as expressed in Eq. (15):

$$h_t = o_t * tanh(C_t) \tag{15}$$

The input data is $x = \{x_1, x_2, x_3, ..., x_n\}$, and the output data is $y = \{y_1, y_2, y_3, ..., y_n\}$, using $E_t$ and $D_t$ to denote the hidden states of the encoder and decoder, and $LSTM_{en}$ and $LSTM_{dn}$ to denote the processing units used in the encoder-decoder network in the Seq2Seq model. The inputs and outputs are represented by Eqs. (16) and by (17):

$$E_t = LSTM_{en}(x_t, E_{t-1}) \tag{16}$$

$$D_t = LSTM_{dn}(x_t, D_{t-1}) \tag{17}$$

The output layer is a *SoftMax* activation function that normalizes $K$ real numbers to obtain $K$ probability distributions. After the *SoftMax* activation function, each component is mapped in the (0, 1) interval and all components sum to 1. The output is mapped into the probability distribution of the predicted output class. Setting $Z = (Z_1, Z_2, ..., Z_n) \in R^K$, the standard function is defined as represented by Eq. (18):

$$\text{softmax}(Z_j) = \frac{e^{z_j}}{\sum_{K=1}^{K} e^{z_k}}, j = 1, ...., k \tag{18}$$

The labels obtained from the prediction are returned by the *argmax* functions, and the final predicted classification result is Y, represented by Eq. (19):

$$Y = \text{argmax}\left[\text{softmax}(z_j)\right] \tag{19}$$

## Intelligent controller module

### Before accessing the user connection

The connection initiation component submits the access request to the intelligent controller through the agent, and the intelligent controller receives the access request and predicts the trust value of the user through UBTA, in which the trust assessment engine module continuously calculates the "trust value" of the terminal based on the input of external historical behavior data to achieve continuous dynamic authentication, and at the same time, it also needs to The security status of the accessed user is updated with weights to analyze the current behavior.

According to its user prediction value greater than 0.5, the intelligent controller grants the corresponding permission for the operation threshold according to its specific value, and at the same time sends the flow table to the policy enforcement component, which transmits the latest rules to the gateway, which enables the terminal to directly access the protected resources through the two-way encryption channel MTLS. If the user prediction value is less than 0.5, the intelligent controller will not respond directly.

### After accessing the user connection

Throughout the flow table interaction, the traffic anomaly detection model CALSeq2Seq in the user behavior analysis engine module also monitors the traffic in real-time to prevent malicious attacks. In order to update the flow table in a timely manner, the intelligent controller also issues each flow table to control whether it still wants to maintain the

communication while predicting, and each switch updates the IP address and port number of the packet to check whether the Each switch updates the packet's IP address and port number to check whether there are matching fields in the flow table. Flows are logged only if a match is found; if no match is found in the switch flow table, the packet is discarded. And the attacked port is blocked for 120 s before it is opened again, and the blocking time increases dynamically with the frequency of each attacker stopping the attack completely.

## Communication execution module

This module mainly includes three main components: connection initiation component, connection acceptance component, and policy enforcement component.

(1) Connection initiation component: This component mainly consists of the user and agent, the user is the subject of access, and it is the identity mapping of real users in the network. The agent is the device used to initiate access, the system environment hardware and software, or the program code to initiate access.

(2) Connection acceptance component: This component mainly consists of the gateway and the protected resources. The gateway takes measures to allow or block the communication links connected by the user in real time through the policies sent from the policy enforcement component. Protected resources are important information hidden throughout the network, and only users who pass authentication and detection can access the resources.

(3) Policy enforcement component: This component is responsible for enabling, monitoring, and eventually ending the connection between the accessing subject and the enterprise resource. The policy enforcement component communicates with the intelligent controller to forward requests and receive real-time updates to policies. It controls the communication link between two different connected components.

## SIMULATION EXPERIMENTS

In order to implement the proposed intelligent zero-trust security framework IZTSDN, the core logical components of the software-defined boundary SDP are integrated in the SDN network simulator Mininet, and the simulation platform MiniIZTA, which can support zero-trust technology and deep learning algorithms, is built. The core logic components of SDP: intelligent controller, policy enforcement component, connection initiation component, and connection acceptance component are integrated into Mininet to build the MiniIZTA emulation platform.

## MiniIZTA emulation platform initialization

To test the usability of the Mininet platform with respect to the SDN emulation environment, a basic network topology was first created that included the attacker, the legitimate user and the service resource the SDN network topology was set up graphically by selecting the icons in the toolbar and continuing to configure the basic properties of each device throughout the network environment. After all the initialization configuration is completed, open the command line interface of the platform to test whether the whole network can communicate normally through the ping command, each terminal can ping

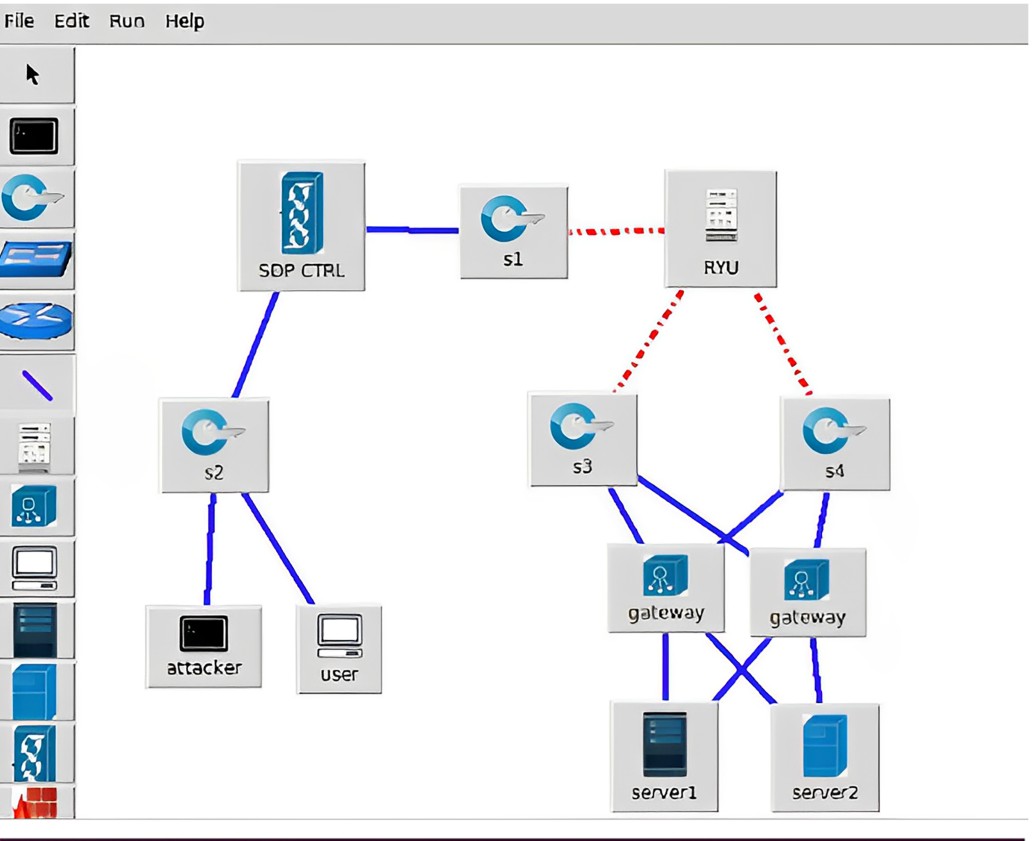

**Figure 4** MiniIZTA platform.

the other, indicating that the routing configuration policy usually is accessible to each other during the initialization state. The MiniIZTA Platform is given in Fig. 4.

SDP core components will be integrated to initialize the MiniIZTA platform, setting both attackers and legitimate users as connection initiation components, while policy enforcement components are set in front of service resources to forward flow table commands from the controller, and access resources are set as connection acceptance components, connecting them to the SDN switch to form a basic SDN network environment.

## Integrating zero trust components

Integrate the software-defined perimeter SDP core components in Mininet to implement zero-trust technology. Through the API interface, the traffic anomaly detection model

**Table 4 Detailed settings.**

| Equipment | Software | Specification |
|---|---|---|
| Controller module | – Linux Ubuntu 18.04 | – 16 GB DDR4 |
| | – Open vSwitch 2.5.6 | – 4.20 GHz Inteli7-8650U |
| VM1 (Legal users) | – Linux Ubuntu 16.04 | – 4 GB RAM |
| | – Mininet 2.3.0 | – 40 GB SSD |
| | – fwknop client (IH) | – 2 NICs |
| VM2 (Illegal users) | – Linux Ubuntu 16.04 | – 4 GB RAM |
| | – Mininet 2.3.0 | – 40 GB SSD |
| | – fwknop client (IH) | – 2 NICs |
| VM3 (Gateways/Resources) | – Linux Ubuntu 16.04 | – 4 GB RAM |
| | – RYU 4.34 | – 40 GB SSD |
| | – fwknop server (AH) | – 2 NICs |

CALSeq2Seq and the trust authentication algorithm UBTA are written in the controller to achieve real-time detection of network traffic and dynamic and sustainable fine-grained authentication authorization of identity. Zero Trust technology makes up for the disadvantage of "connect first, verify later" in traditional TCP/IP by performing authentication before connection establishment. At the same time, traditional scanning tools cannot scan ports because there are no open ports, thus further securing the network. The entire simulation platform MiniIZTA is launched through both the command line and graphical interface, and the network topology is created after the initialization state is verified and connected.

The intelligent zero-trust security framework IZTSDN writes the traffic anomaly detection model CALSeq2Seq and the trust authentication algorithm UBTA into the controller to implement the corresponding functions. to resist attacks effectively.

## Setup of the MiniIZTA emulation platform

This MiniIZTA platform consists of four Linux Ubuntu virtual machines, where the user module is installed on VM1 (legal user) and VM2 (illegal user), the controller module is installed on the host, and the gateway and resource modules are installed on VM3 virtual machine. The switches in the simulation environment are SDN switches supporting the OpenFlow protocol, and the controller is using Ryu. The detailed specification of the VM setup. Detailed settings are given in Table 4.

Before testing the network emulation attack, the Ryu controller needs to be started, and the process can be divided into initializing the configuration, creating an application instance, connecting to the switch, starting the message loop, loading the application, and running the application. The Ryu controller first loads its default configuration file ryu. conf, and second, uses the logging module in the Python standard library to log logging information. At startup, the controller initializes the module and determines the logging level and output based on the settings in the configuration file. Then, it creates an application instance and registers the zero-trust SDP core component code and the deep

```
mininet@ubuntu:~/SDN-DDOS-Detection$ ryu-manager controller.py
loading app controller.py
loading app ryu.controller.ofp_handler
instantiating app ryu.controller.ofp_handler of OFPHandler
instantiating app controller.py of SimpleSwitch13
['03/07/2022, 17:55:09', '3', '0', '0.0']
['03/07/2022, 17:55:09', '-2', '0', '1.0']
```

**Figure 5  Ryu controller starts normally.**

learning algorithm model into the framework to detect abnormal traffic. Immediately afterward, a connection needs to be established with the OpenFlow switch for message interaction. The controller will listen to a TCP port and wait for a connection request from the switch. Once the connection is successful, the controller sends a Hello message for negotiation and then enters the normal message processing flow. The final stage is to load all the modules in the directory and start running all the executable applications, implementing all the functions of the IZTSDN security framework regarding the Intelligent Engine module on the Ryu controller, in addition to the other component contents that are programmatically defined in the corresponding terminal software to implement their functions. The Ryu controller start-up interface, which starts communicating with other components in the simulation platform. Ryu controller starts normally is given in Fig. 5.

## Port scan simulation attacks

Nmap is a popular open-source network discovery and security auditing tool that can be used to scan hosts and services on a computer network and generate detailed reports. It uses different techniques to identify information such as the operating system, application version, and open ports running on the target host. The specific steps of the attack test are as follows:

(1) Open a command line terminal window and on the command line, enter the IP address or domain name of the target to be scanned.

(2) Determine the object to be scanned and the port number. The scan will scan 1,000 common ports by default, and if more ports need to be scanned, you need to specify the port range. Experiment by testing to scan port 22, which is the default port for SSH protocol, an encrypted network protocol used for secure remote login and other network services on insecure networks.

(3) Wait and check the scan result, a normal scan of port 22 using Nmap returns a status of open, which means that the port is open and there is a program running that is listening to the port, which indicates that services associated with the port may be accessible and exploitable. When the IZTSDN security framework was used, port 22 was scanned again with Nmap, and the status returned was filtered, which means that the port did not explicitly respond to the probe packets sent by Nmap. This is because IZTSDN does not give any response information until the identity of the accessing user is confirmed, which fundamentally prevents the transmission of similar probe packets.

```
loading app controller.py
loading app ryu.controller.ofp_handler
instantiating app controller.py of SimpleSwitch13
instantiating app ryu.controller.ofp_handler of OFPHandler
['03/07/2022, 18:26:24', '544', '543', '0.0']
['03/07/2022, 18:26:27', '280', '280', '0.0']
['03/07/2022, 18:26:30', '180', '180', '0.0']
['03/07/2022, 18:26:33', '190', '190', '0.0']
['03/07/2022, 18:26:37', '189', '189', '0.0']
['03/07/2022, 18:26:39', '189', '189', '0.0']
['03/07/2022, 18:26:43', '190', '190', '0.0']
['03/07/2022, 18:26:46', '190', '190', '0.0']
['03/07/2022, 18:26:49', '106', '106', '0.0']
['03/07/2022, 18:26:52', '144', '144', '0.0']
['03/07/2022, 18:26:55', '144', '144', '0.0']
```

**Figure 6** **DDoS simulation attack test generation traffic.**

## DDoS simulation attacks

In the DDoS simulation attack experiment, the attack test is launched using hping3, which is a common network tool that can be used to generate various types of traffic packets, including normal traffic and attack traffic. In hping3, normal and attack traffic can be simulated by setting different parameters and options. DDoS simulation attack test generation traffic is given in Fig. 6.

To simulate normal traffic, use the "-S" option to send TCP SYN packets, which is the first step of the TCP three-times handshake, indicating a connection request. Use the "-1" option to send ICMP Echo request packets, also known as Ping request packets. Use the "-2" option to send UDP packets, which are the type of packets that a normal application would send.

When simulating attack traffic, use the "–flood" option to send a TCP SYN Flood attack, which sends a large number of TCP SYN packets to the target host, making its service unavailable. Use the "–udp" option to send a UDP Flood attack, which sends a large number of UDP packets to the target host. Use "–icmp" option to send an ICMP Echo Flood attack, it will send a large number of Ping request packets to the target host, these attacks will cause the target host to run out of resources, but also are some common denial of service attacks.

## PERFORMANCE EVALUATION

In this section, we implement the IZTSDN security framework on the simulation platform and evaluate its performance, mainly from the two evaluation metrics of network performance and traffic anomaly detection accuracy to verify the security of this framework.

### Network performance evaluation

Two virtual machines were set up in the experiment, VM1 as a legitimate user client and VM2 as an illegitimate user client, with the aim of testing the impact of access under normal conditions and access under DDoS attacks on network performance. Initialize server-side bandwidth is given in Fig. 7.

```
mininet@ubuntu:~$ sudo iperf -s -i 3
------------------------------------------------------------
Server listening on TCP port 5001
TCP window size:  128 KByte (default)
------------------------------------------------------------
[  4] local 192.168.41.174 port 5001 connected with 192.168.41.177 port 37574
[ ID] Interval       Transfer     Bandwidth
[  4]  0.0- 3.0 sec  1.17 GBytes  3.34 Gbits/sec
[  4]  3.0- 6.0 sec  1.26 GBytes  3.60 Gbits/sec
```

**Figure 7** Initialize server-side bandwidth.

The initial bandwidth of the network environment. The default port is set to 5,001 using TCP protocol, and the initial TCP bandwidth is about 3.5 Gbps, while the UDP bandwidth is set to 10 Mbps to verify the network throughput under normal conditions and after using the IZTSDN security framework. The illegal user client VM2 uses the hping3 tool to launch DDoS attacks and port scanning attacks to destroy the throughput in the whole network environment by consuming a large amount of bandwidth of the SDN network. Also, the Iperf tool is used to display the network throughput in real-time in both cases when launching the attack test. Note that hping3 can construct virtual TCP packets in a short period of time to massively consume the bandwidth of the SDN network.

On VM2, the hping3 tool is used to launch attacks, by default in TCP mode. hping3 spoofs random IP addresses to quickly launch TCP requests, thus consuming bandwidth. In Iperf test results, "bandwidth" usually refers to the actual data transfer rate, which is also known as network throughput. It reflects the actual efficiency of data transmission during the test period. The overall performance of the entire network drops dramatically to around a few Mb/s or even lower. Launch DDoS and PS attacks under normal circumstances is given in Fig. 8. This has a significant impact on normal access to services for legitimate customers. On the contrary, with the security framework IZTSDN turned on, the throughput of the SDN network environment was hardly affected significantly and could basically be maintained at around 2.81 Gb/s. Launch DDoS and PS attacks under IZTSDN protection is given in Fig. 9. This indicates that for unauthorized DDoS attacks, IZTSDN, an intelligent zero-trust security framework, can defend well against such unauthorized network attacks.

Under normal circumstances, when DDoS attacks and PS attacks are launched, the data volume and bandwidth of the entire network environment transmitted within a certain time interval drop sharply to less than 1%, and it is no longer possible to provide normal services to normal users. In contrast, after using the security framework IZTSDN, the same scale of the attack is launched and the network throughput is observed. IZTSDN is basically free from unauthorized attacks and can forward and receive packets normally within the 90 s of the test time. Change in network throughput in the 90 s is given in Fig. 10. The average network throughput counted can also reach 2.81 Gb/s, and the amount of transmitted data is also basically the same as when it was not subject to the original attack, and the availability rate can reach about 80.5%.

The Wireshark tool was used to analyze the traffic data during the entire network environment communication. When the normal situation is unprotected, it can be seen

```
IZTA@ubuntu:~$ sudo iperf -c 192.168.41.174 -i 3 -t 30
------------------------------------------------------------
Client connecting to 192.168.41.174, TCP port 5001
TCP window size: 85.0 KByte (default)
------------------------------------------------------------
[  3] local 192.168.41.177 port 53998 connected with 192.168.41.174 port 5001
[ ID] Interval       Transfer     Bandwidth
[  3]  0.0- 3.0 sec  1.38 MBytes  3.84 Mbits/sec
[  3]  3.0- 6.0 sec  1.25 MBytes  3.50 Mbits/sec
[  3]  6.0- 9.0 sec  1.25 MBytes  3.50 Mbits/sec
[  3]  9.0-12.0 sec   896 KBytes  2.45 Mbits/sec
[  3] 12.0-15.0 sec  1.25 MBytes  3.50 Mbits/sec
```

**Figure 8  Launch DDoS and PS attacks under normal circumstances.**

```
IZTA@ubuntu:~$ sudo iperf -c 192.168.41.174 -i 3 -t 30
------------------------------------------------------------
Client connecting to 192.168.41.174, TCP port 5001
TCP window size: 85.0 KByte (default)
------------------------------------------------------------
[  3] local 192.168.41.178 port 40212 connected with 192.168.41.174 port 5001
[ ID] Interval       Transfer     Bandwidth
[  3]  0.0- 3.0 sec  1.02 GBytes  2.93 Gbits/sec
[  3]  3.0- 6.0 sec   842 MBytes  2.36 Gbits/sec
[  3]  6.0- 9.0 sec  1.02 GBytes  2.92 Gbits/sec
[  3]  9.0-12.0 sec   980 MBytes  2.74 Gbits/sec
[  3] 12.0-15.0 sec  1.09 GBytes  3.11 Gbits/sec
```

**Figure 9  Launch DDoS and PS attacks under IZTSDN protection.**

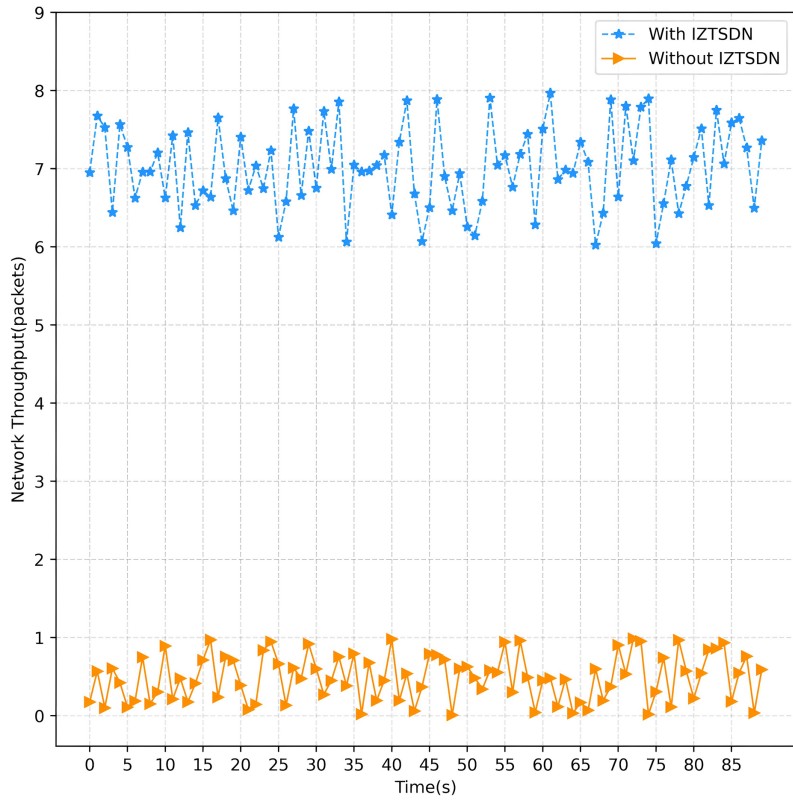

**Figure 10  Change in network throughput in the 90 s.**

**Table 5 Average throughput.**

| Flow type | Without IZTSDN | With IZTSDN |
|-----------|----------------|-------------|
| TCP | 3.36 Mb/s | 2.81 Gb/s |
| UDP | 1.7 Mb/s | 1.02 Gb/s |

that almost no packets can be sent or received normally, while when the IZTSDN security framework is on, the network throughput and bandwidth are almost unaffected. This is because unauthorized network attacks simply cannot be authorized by the intelligent controller's authentication and will not respond to data requests that are illegally accessed, nor will the server respond to the requests it sends. Port scanning attacks also give any response to those access requests that are not authenticated when the IZTSDN security framework is on, which makes it impossible for attackers to scan any open ports and get any valid information.

In an ideal situation, IZTSDN would block all unauthenticated traffic, meaning that the entire network should be used for authenticated clients. However, in the actual network environment tested, there is also some lesser inflow of unauthorized traffic for a short period of time when legitimate clients first start to pass through, so the attack consumes some bandwidth, but with a smaller drop.

It is observed in this experiment that the bandwidth available for TCP is larger than the bandwidth available for UDP, analyzing that during TCP communication three handshakes need to be established to acknowledge the data it transmits, while UDP does not need to wait for acknowledgment and will forward accepted data faster than TCP. The final test results. Average throughput are given in Table 5.

## Evaluation of traffic anomaly detection

### Experimental environment and data set introduction

The experimental platform is only used to build the neural network of the model with Keras, using Python as the main programming language, and using the operating system of the host Windows 10 64-bit workstation, whose main hardware and software environment configuration. The main hardware and software environment configuration are given in Table 6.

Since 1998, the relevant cybersecurity agencies have evaluated the existing datasets and found that most of them are outdated and cannot meet the complex and diverse network environments of today. Some of the datasets were collected in traditional network environments and could not be applied to SDN network environments, and another part of the datasets contained attack types that no longer encompassed some of the existing publicly available attack types.

The publicly available dataset provided by Bennett University Research Institute is used by simulating a collection of benign TCP, UDP, and ICMP traffic as well as malicious

**Table 6 Main hardware and software environment configuration table.**

| Software and hardware | Models and parameters |
| --- | --- |
| Deep learning frameworks | Keras2.4.3 |
| Python | 3.8 |
| Anaconda | V4.12 |
| CPU | Intel Core i9-9700K |
| Running memory | 64 GB Kingston |
| Solid state drive | 1TB 980 SAMSUNG |
| Video card | NVIDIA GeForce GTX2080 |

traffic runs (including TCP Syn attacks, UDP Flood attacks, ICMP attacks, and other attacks).

The entire dataset contains more than 100,000 rows of flow table data for training and testing the model. For feature selection, a 1D CNN was used to extract effective flow features from the originally collected data. When training the model, the dataset was randomly segmented for cross-validation, and several evaluation metrics were used to assess the performance of the model, such as accuracy, precision, recall, and F1 score. Through these measures, the dataset can be effectively used to improve the performance and robustness of the model. Key features included in the dataset are given in Table 7.

### Model training and optimization

After several times of model training and optimization, the experiments set the optimizer as Adam, the loss function as binary_crossentropy, the number of data samples captured in one training (Batch_sizes) set to 32, the initial learning rate set to 0.01, and the Epochs set to 120, where the Adam optimizer combines the advantages of both AdaGrad and RMSProp optimization algorithms, introduces gradient momentum and gradient squared momentum, and makes the learning rate close to the optimal state by automatically adjusting the learning rate, and also introduces Dropout to prevent the problem of overfitting during model training, setting it to 0.2. During the experiments, in the SDN dataset of 104,345 data, there are still 76,009 data in the preprocessing module, and 60,807 of them are extracted as the data set to train the model, and 15,202 data are used as the test set to test the model.

In the feature extraction module, convolutional layers is 2, number of filters is 32, kernel size is 3*3, stride is 1, pooling layer is 2*2, activation function is ReLU.

In the detection classification module, number of LSTM hidden units is 64, number of stacked LSTM layers is 1, learning rate is 0.01, epochs is 120, Batch_sizes is 32, loss function is binary cross entropy, optimizer is Adam, dropout is 0.2.

Under the same public SDN dataset, a variety of deep learning detection algorithms are compared, and the experimental results show that the CALSeq2Seq model has the best overall performance and can achieve good detection results in the simulated environment,

**Table 7 Key features included in the dataset.**

| Feature name | Meaning |
|---|---|
| DT | Date and time of the data |
| Switch | Data path id in the topology |
| Src (Source IP) | Source IP address |
| Dst (Destination IP) | Destination IP address |
| Packet count | Number of packets sent in the stream |
| Byte count | Number of bytes sent in the stream |
| Duration | Time of the stream in the switch |
| Duration_nsec | Time of the stream in the switch in nanoseconds) |
| Total_duration | Sum of dur_sec and dur_nsec |
| Flows | Which switch the stream passed through |
| Packetins | Number of packets transmitted to the controller |
| Pktperflow | Number of packets for a single stream |
| Byteperflow | Number of bytes in the stream (byte count during the stream) |
| Pktrate | Packet rate (number of packets sent per second) |
| packetlistener | Incoming traffic packet listener |
| Protocol | Protocol |
| Port_no | Port number of the switch to which the request was sent |
| Tx_bytes | Number of bytes sent on the switch port |
| Rx_bytes | Number of bytes transmitted to the switch port |
| Tx_kbps | Kilobytes transmitted per second |
| Rx_kbps | Kilobytes received per second |
| Total_kbps | The bandwidth of the switch port |
| Label | Classification labels (0 normal, 1 malicious) |

and the CALSeq2Seq model has converged in about 120 rounds, and the accuracy and loss rates have stabilized. CALSeq2Seq model accuracy and loss rate is given in Figs. 11 and 12. The performance of the proposed CALSeq2Seq traffic anomaly detection model in this article is faster than other models in terms of training speed, optimal fit and accuracy, and low loss rate, which can provide continuous and accurate detection of user behavior monitoring and identification, combined with effective security access control policies to help SDN networks can prevent unauthorized attacks from outside and internal malicious user-initiated network attacks in advance, which can guarantee the security of the network and meet the security requirements of SDN.

### Analysis of experimental results

In classification experiments, samples can be classified as true positives (TP), false negatives (FN), false positives (FP), and true negatives (TN). TP indicates the proportion of attack samples predicted as attack samples, FN indicates the proportion of normal samples predicted as attack samples, FP indicates the proportion of attack samples predicted as normal samples and TN indicates the proportion of normal samples predicted as normal samples. The confusion matrix is usually used to calculate the above

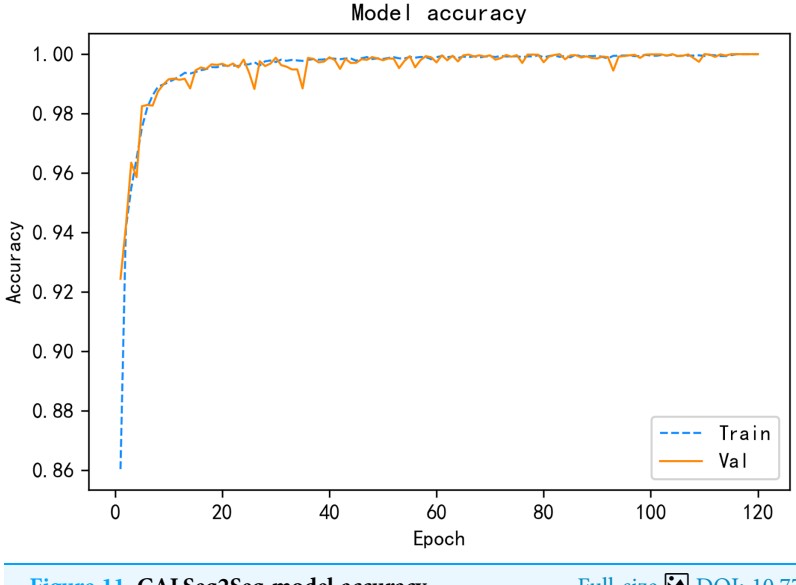

**Figure 11  CALSeq2Seq model accuracy.**

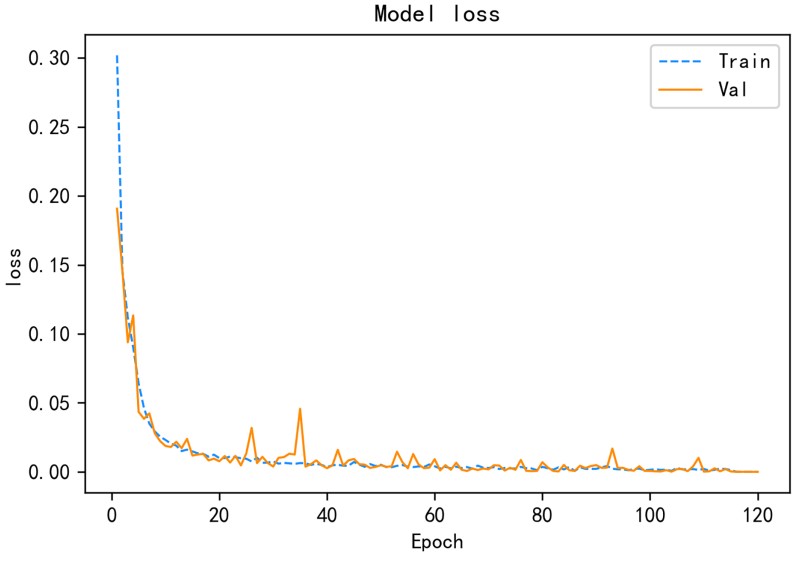

**Figure 12  CALSeq2Seq model loss rate.**

performance parameters. In the binary classification problem, it is defined as a $2 \times 2$ matrix showing the actual and predicted values of the classifier.

From the experimental process of model training and optimization, it can be seen that the traffic anomaly detection model CALSeq2Seq has shown good results in terms of accuracy and loss rate on the existing publicly available SDN dataset. In the experiments, each comparison algorithm will be evaluated using the mentioned datasets, and later also compared with the research knots of other articles for comparative analysis, the models will be compared based on their accuracy, false alarm rate, precision, recall, and

**Table 8  Model performance evaluation.**

| Model | Accuracy | Precision | Recall | False-alarm | F1-measure |
|---|---|---|---|---|---|
| LR | 83.69% | 83.31% | 82.46% | 17.51% | 82.26% |
| SVM | 85.83% | 85.79% | 87.46% | 12.62% | 86.61% |
| KNN | 95.22% | 96.83% | 93.47% | 5.61 | 95.58% |
| RF | 97.2% | 96.56% | 95.45% | 4.52% | 96.23% |
| ANN | 98.2% | 97.43% | 97.84% | 2.53% | 97.12% |
| SVM-RF | 98.8% | 98.27% | 97.91% | 2.17% | 97.65% |
| CNN | 98.74% | 98.75% | 98.9% | 1.98% | 98.6% |
| LSTM | 95.6% | 96.2% | 95.6% | 2.31% | 95% |
| CALSeq2Seq | 99.56% | 99.62% | 98.89% | 1.22% | 99.17% |

F1-measure, which are the main evaluation metrics for performance. The formula is shown by Eqs. (20)–(24).

$$Accuracy = \frac{TP + TN}{TP + TN + FN + FP} \tag{20}$$

$$False\ Alarm\ Rate = \frac{FP}{TN + FP} \tag{21}$$

$$Precision = \frac{TP}{TP + FP} \tag{22}$$

$$Recall = \frac{TP}{TP + FN} \tag{23}$$

$$F1 - measure = \frac{2 precision \times recall}{precision + recall} \tag{24}$$

The publicly available SDN dataset was used, and the model performance was comprehensively evaluated by calculating the evaluation metrics of accuracy, false alarm rate, precision, recall, and F1-measure. The CALSeq2Seq model was compared with other deep learning models. It can be directly observed from the table that the CALSeq2Seq model achieves the highest scores in all evaluation metrics, showing excellent performance. Model Performance Evaluation are given in Table 8.

The CALSeq2Seq model achieves more than 98% accuracy with traditional machine learning models as well as in other deep learning models with ANN, CNN, and SVM-RF, which greatly improves the efficiency of network resource utilization and reduces network latency. In terms of false alarm rate, the CALSeq2Seq model also has a strong anti-noise capability, which is significantly lower than all other models, further demonstrating the reliability of CALSeq2Seq. In terms of both accuracy rate and recall rate, the CALSeq2Seq model performs equally well with SVM-RF and CNN, especially in terms of recall rate, CALSeq2Seq model can reach more than 98%, which significantly improves the coverage of the SDN network. The F1-measure, a comprehensive evaluation metric, shows the superior performance of the CALSeq2Seq model in recognition and classification tasks.

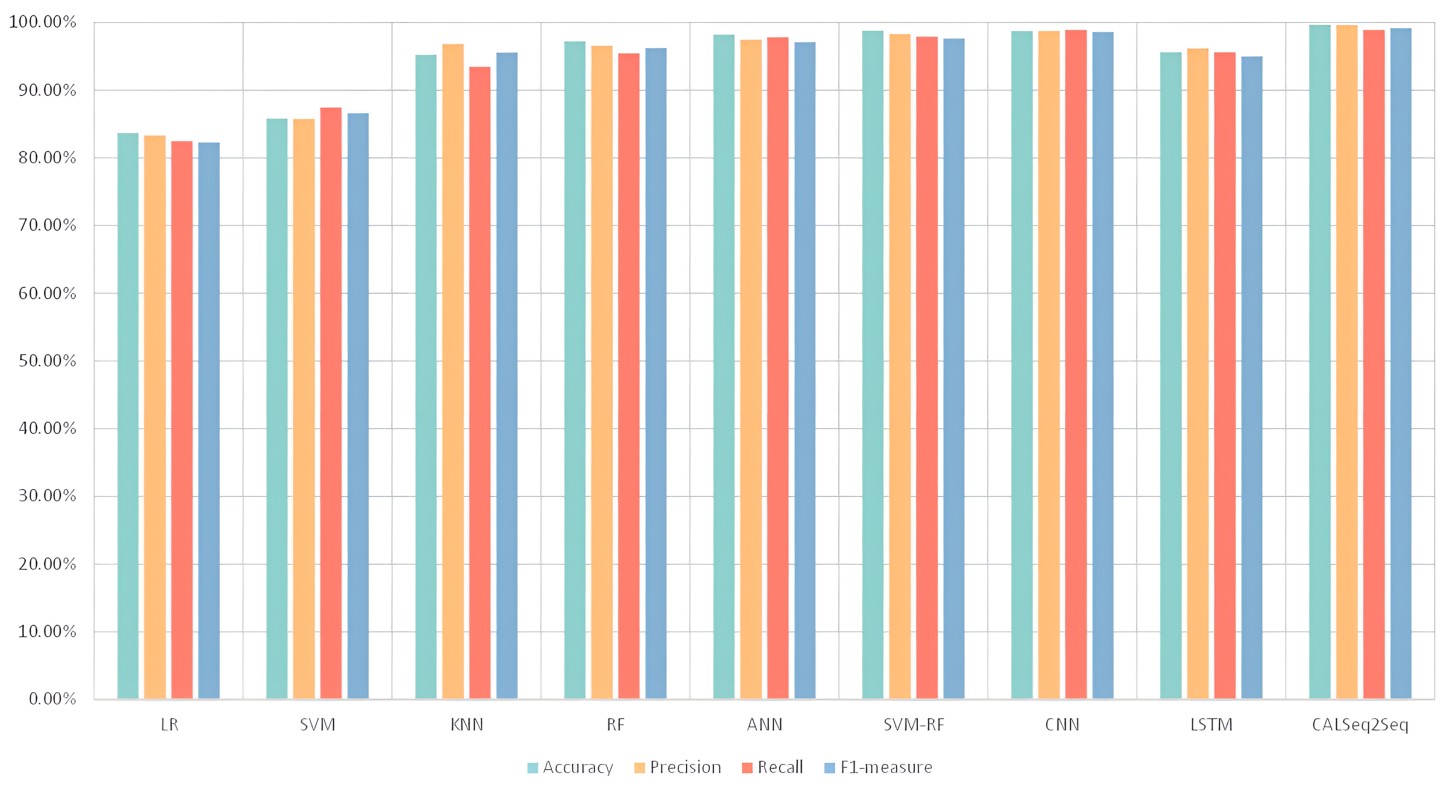

**Figure 13 Performance evaluation.**

In the experimental analysis, the researchers used histograms to demonstrate the performance of the model in the four evaluation metrics of accuracy, precision, recall, and F1-measure. Performance evaluation is given in Fig. 13. From the histogram, it can be seen that the CALSeq2Seq model performs very well balanced and excellent in these four metrics. This means that the model is able to consider these metrics at the same time and does not perform too well on one metric and causing the other metrics to drop. Therefore, using the CALSeq2Seq model can better detect malicious traffic attacks in SDN.

Figure 14 is the ROC plot reflecting the relationship between sensitivity and specificity. In this figure, the abscissa X-axis represents the specificity, also known as the false positive rate or false positive rate. As the X-axis is closer to zero, the accuracy is higher. The Y-axis is the sensitivity, also known as the true positive rate or sensitivity. The higher the Y-axis, the better the accuracy. The closer the curve is to the top-left corner (the smaller the X is and the larger the Y is), the higher the prediction accuracy will be, and you can intuitively see that the classifier near the top-left corner has the highest accuracy.

CALSeq2Seq exhibits the highest sensitivity and the lowest false alarm rate, which means that this algorithm has the highest classifier accuracy. Therefore, when evaluating the pros and cons of the model algorithm, it is analyzed according to the ROC curve to judge the performance of the algorithm more accurately.

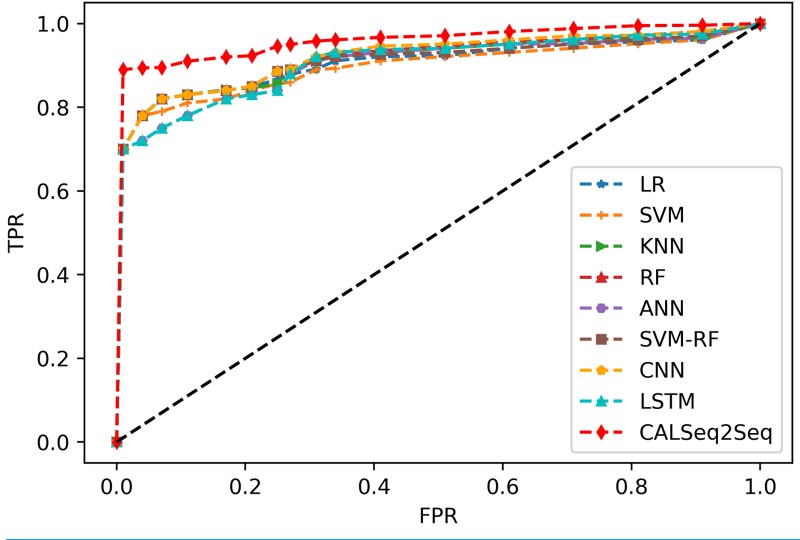

**Figure 14 ROC.**

Literature research found that some researchers claim to use SDN networks in their articles, but use the classical datasets KDD Cup and NSL-KDD under traditional networks for their experiments. However, these datasets are not fully applicable to SDN network scenarios. In addition, some authors did not disclose the datasets used but tested them by building their own network environments, an approach that may run the risk of lacking effective validation.

Currently, there are very limited publicly available SDN datasets. Therefore, the publicly available SDN dataset was specifically chosen to ensure the reliability of this article study. This dataset contains more than 100,000 data samples, all generated and collected in the SDN network environment, which can more accurately reflect the characteristics of SDN networks. This allows the model to be constructed based on this dataset and deployed into the proposed security framework IZTSDN, thus making the research in this article more credible and reproducible. Compared with the algorithmic models proposed by other researchers, the model performance comparison of different solutions are given in Table 9. The experiments show that the CALSeq2Seq model proposed in this article has the highest accuracy (*Ye et al., 2018*; *Ahuja et al., 2021*; *ElSayed et al., 2021*; *Sanagavarapu & Sridhar, 2021*).

The network attack test generates normal and attack traffic by hping3 simulation, and the parameters of CALSeq2Seq, the anomaly traffic detection model in the IZTSDN security framework, are tuned several times to achieve better results. After the entire simulation platform is initialized, the Ryu controller is started and all other components are opened. IZTSDN performs accurate anomaly detection on user traffic that has been authorized by authentication, and if suspicious malicious traffic is found, the controller issues a flow table forwarding policy to make the policy enforcement component disconnect the data channel established between the user and the resource. The blocking time increases dynamically with the frequency of each attacker for 120 s to completely stop the attack.

**Table 9 Model performance comparison of different solutions.**

| Author | Model | Network type | Data set | Accuracy |
|---|---|---|---|---|
| ElSayed | CNN+RF | SDN | InSDN (Not public) | 97% |
| Sanagavarapu | ANN+LSTM | SDN | Customization (Not public) | 98.87% |
| Ye | SVM | SDN | Customization (Not public) | 95.24% |
| Ahuja | SVC-RF | SDN | SDN (Public) | 98.8% |
| Self | CALSeq2Seq | SDN | SDN (Public) | 99.56% |

**Table 10 Performance comparison of different schemes.**

| Scheme | Throughput (Zero trust) | Accuracy (Deep learning) |
|---|---|---|
| Self: IZTSDN | 2.81 Gb/s (80.5%) | 99.26% |
| Han: OverWatch | Not supported | 96% |
| Moubayed: SDP | 1.51 Gb/s (75.5%) | Not supported |

The traffic anomaly detection model CALSeq2Seq is applied to the control of IZTSDN, an intelligent zero-trust security framework, and is used to detect user traffic in the network in real time. In the network attack test conducted in the simulation platform MiniIZTA, the experimental results show that CALSeq2Seq has an accuracy of 99.26% on the test set. In addition, the average score after five-fold cross-validation is 0.9975, further verifying that CALSeq2Seq's performance is optimal. The experimental results show that CALSeq2Seq is a very reliable traffic anomaly detection model that can play an important role in SDN network environment.

In the comparative analysis of schemes in Table 10, this article combines zero trust and deep learning models to propose an intelligent zero trust security framework IZTSDN, which has better performance than using any technology alone. The OverWatch framework proposed by *Han et al. (2018)* does not support zero trust, and its throughput is not tested in the article, only mentioning that its communication overhead is small. In addition, although the coordinated attack detection mechanism is proposed in this article, the experimental results show that the detection accuracy is only about 96%, which is far lower than the accuracy of the model proposed in this article. The SDP framework proposed by *Moubayed, Refaey & Shami (2019)* mainly utilizes the zero-trust "verify first, then connect" feature, and finds about 75.5% throughput in the experimental attack test. However, the framework does not support deep learning algorithms to achieve fine-grained detection.

## CONCLUSIONS

In this article, we propose an intelligent zero-trust security framework named IZTSDN for securing network resources in SDN-based networks by combining deep learning with zero-trust architecture. The framework comprises five core modules: a data collection module, a trust assessment engine module, a user behavior analysis engine module, an

intelligent controller module, and a communication execution module. To detect attack traffic in SDN networks, we introduce a traffic anomaly detection model called CALSeq2Seq based on deep learning algorithms. The model employs a one-dimensional convolutional neural network (1DCNN) to extract network features and introduces a self-attention mechanism to achieve a fusion of key features. Additionally, the LSTM-Seq2Seq structural network is used as a codec to learn the dependencies between features. Finally, the prediction results are output through the fully connected layer. Experimental evaluations using publicly available SDN datasets show that the CALSeq2Seq model outperforms other algorithms. We also address the issues associated with the traditional "connect first, authenticate later" access model in SDN networks by proposing a trust algorithm called UBTA based on zero-trust technology. The UBTA algorithm uses the user's historical behavioral data to evaluate its trust value and grants corresponding authority based on its given operation threshold, enabling a dynamic authorization process. To validate our proposed intelligent zero-trust security framework IZTSDN, we build the simulation platform MiniIZTA by integrating the zero-trust component extension Mininet network tool. We implement the IZTSDN security framework proposed with the help of the MiniIZTA simulation platform and verify its usability and security by designing network simulation attack tests and security tests. In the experiments, we use two evaluation metrics, throughput, and traffic anomaly detection accuracy, to evaluate the performance of the IZTSDN security framework. The experimental results show that the IZTSDN security framework can maintain about 80.5% throughput when encountering unauthorized DDoS attacks and port scanning attacks. Furthermore, the traffic anomaly detection model CALSeq2Seq has an accuracy rate of 99.26% in the test, with an average score of 0.9975 after five-fold cross-validation, further validating its performance in the simulation test. Additionally, the security framework performs port blocking of the attacker's IP to prevent further malicious attacks.

## DISCUSSION

With the continuous development of technology, there is still room for further improvement in this area. Therefore, based on the existing article, future SDN security issues are further studied as follows:

(1) The traffic anomaly detection model proposed in this article, although can greatly improve the accuracy of network anomaly traffic detection and reduce the false alarm rate. However, it is not enough to identify the type of network traffic attack alone, and the addition of mitigation and traceability mechanisms can be considered later.

(2) The network traffic used in this study is unencrypted. Nowadays, some of the attack traffic is transmitted after encryption, which is harmful, long attack cycle time, is concealment, and is not easy to be detected. Therefore, the predictive identification of encrypted traffic needs to be added at a later stage.

(3) The method of trust algorithm proposed in this article needs to be further optimized. This is due to the nature of the trust itself such as interactivity which makes it difficult to be quantified, and how to further derive the trust value of the access subject still needs a lot of theoretical research.

For the mitigation of network traffic attacks, in the short term, we can try to use some security policies, and in the long term, we can develop a set of security protocols for specific scenarios to mitigate abnormal traffic attacks. As for traceability techniques, one can consider adding a threat model to trace the attacker's attack path to prevent such attacks in advance at a later stage. For the identification and processing of encrypted traffic maybe consider transforming it into images or text and using neural networks to predict the identification. The optimization of trust algorithms is more reflected in the algorithm, perhaps in the future, we can try to add some security algorithms to ensure the degree of trust.

### Funding
This work was supported by the National Natural Science Foundation of China (Grant Nos. 61461027), the Gansu Provincial Science and Technology Program Fund (20JR5RA467), and the Lanzhou University of Technology Graduate Program. The funders had no role in study design, data collection and analysis, decision to publish, or preparation of the manuscript.

### Grant Disclosures
The following grant information was disclosed by the authors:
National Natural Science Foundation of China: 61461027.
Gansu Provincial Science and Technology Program Fund: 20JR5RA467.
Lanzhou University of Technology Graduate Program.

### Competing Interests
The authors declare that they have no competing interests.

### Author Contributions
- Xian Guo conceived and designed the experiments, performed the experiments, analyzed the data, performed the computation work, prepared figures and/or tables, authored or reviewed drafts of the article, and approved the final draft.
- Hongbo Xian conceived and designed the experiments, performed the experiments, analyzed the data, performed the computation work, prepared figures and/or tables, authored or reviewed drafts of the article, and approved the final draft.
- Tao Feng conceived and designed the experiments, prepared figures and/or tables, and approved the final draft.
- Yongbo Jiang performed the experiments, prepared figures and/or tables, and approved the final draft.
- Di Zhang analyzed the data, prepared figures and/or tables, and approved the final draft.
- Junli Fang performed the computation work, authored or reviewed drafts of the article, and approved the final draft.

## Data Availability

The code is available at GitHub and Zenodo:

- https://github.com/Root-xhb/IZTSDN

- Guo, X., Xian, H., Feng, T., Jiang, Y., Zhang, D., & Fang, J. (2023). An intelligent Zero Trust secure framework for Software Defined Networking. In PeerJ Computer Science. Zenodo. https://doi.org/10.5281/zenodo.10005690.

## Supplemental Information

Supplemental information for this article can be found online at http://dx.doi.org/10.7717/peerj-cs.1674#supplemental-information.

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
