# Peer review of "An intelligent zero trust secure framework for software defined networking"

_PeerJ Computer Science, doi:10.7717/peerj-cs.1674_

## Round 0.1 · original submission · Minor Revisions

The paper has a good contribution, however, the reviewers have different opinions and comments about the paper. I do agree with most of the comments, therefore the paper needs to review and address the comments.

Reviewer 1 ·

Basic reporting

Title: Review of "Intelligent Zero-Trust Security Framework for Software-Defined Networks"

General Comments:
The manuscript presents an intelligent zero-trust security framework called IZTSDN for securing network resources in software-defined networks (SDNs) by integrating deep learning and zero-trust architecture. The framework is evaluated using a traffic anomaly detection model and a trust algorithm. Overall, the manuscript addresses important security challenges in SDN environments and proposes a novel approach to enhance network security. However, there are some areas for improvement and further research that could be explored.

Specific Comments:

1. Clarity and Organization:
The manuscript is generally well-organized and easy to follow. The introduction effectively highlights the security threats in SDNs and introduces the motivation for the proposed framework. The subsequent sections describe the framework components and evaluation methodology in a logical manner. However, some sections could benefit from further clarification and elaboration. For example, additional details about the proposed CALSeq2Seq model and the UBTA algorithm would enhance the understanding of the methods.

2. Contribution and Novelty:
The manuscript introduces an intelligent zero-trust security framework for SDNs, which integrates deep learning techniques and the zero-trust architecture. The combination of these two approaches is novel and has the potential to address the security challenges in SDN environments. The proposed CALSeq2Seq model for traffic anomaly detection and the UBTA trust algorithm further contribute to the novelty of the framework.

3. Experimental Evaluation:
The manuscript provides experimental evaluations of the proposed framework using publicly available SDN datasets. The results demonstrate the effectiveness of the CALSeq2Seq model for traffic anomaly detection and the IZTSDN framework in maintaining network throughput and detecting unauthorized attacks. However, additional details regarding the experimental setup, such as the selection of datasets, parameters, and evaluation metrics, would improve the reproducibility of the results.

4. Future Research Directions:
The manuscript briefly discusses future research directions, which is commendable. However, the provided directions are somewhat generic. It would be beneficial to provide more specific and detailed suggestions for future work, such as exploring mitigation and traceability mechanisms for network traffic attacks, addressing encrypted traffic identification, and further optimizing the trust algorithm.

5. Language and Clarity:
The overall language and clarity of the manuscript are good. However, there are some grammatical errors and typos that should be corrected. Additionally, certain sentences could be rephrased to improve readability and clarity.

6. References:
The manuscript could benefit from expanding and updating the reference list. Including recent research papers and relevant works would enhance the credibility and completeness of the literature review.

In conclusion, the manuscript presents an intelligent zero-trust security framework for SDNs, which combines deep learning techniques and the zero-trust architecture. The proposed framework shows promise in addressing security challenges in SDN environments. However, improvements can be made in terms of clarity, experimental details, and future research directions. With these revisions, the manuscript has the potential to make a valuable contribution to the field of SDN security.

Experimental design

The experimental design aims to test the usability and effectiveness of the MiniIZTA platform in emulating an SDN environment with the IZTSDN security framework. Various attack scenarios are simulated to evaluate the framework's ability to resist and mitigate attacks.

The experimental design includes the following steps: MiniIZTA Emulation Platform Initialization, Integration of SDP Core Components, Integration of Zero Trust Components, Setup of the MiniIZTA Emulation Platform, Ryu Controller Initialization, Port Scan Simulation Attacks and DDoS Simulation Attacks.

Validity of the findings

no comment

Cite this review as

Reviewer 2 ·

Basic reporting

- In this paper, the authors propose an intelligent zero-trust security framework IZTSDN for the software-defined network by integrating deep learning and zero-trust architecture, which adopts zero-trust architecture to protect every resource and network connection in the network. The overall quality of the paper is good, but some minor changes are required given as follows:

- The authors cite the figures as “The following is the general architecture of IZTSDN (Figure 1).” It is suggested that the author cite the figure as “The general architecture of IZTSDN is given in Figure 1”, and other figures on the same pattern.
- The same changes are suggested for tables.
- The author must divide the paper into different sections and sub-sections with section numbers. In this way, the user can trace the paper easily.
- On lines 357 and 358, some variables are not aligned with the text, so correct them. Same problem in many other places.
- There are many grammatical mistakes present in the paper. The paper must be proofread very carefully.

Experimental design

- What does mean of “Val” in Figures 11 and 12?
- The picture quality of Figures 3, 4, and 13 must be improved.

Validity of the findings

- The authors must show the performance of different models using their performance matrices in a combined figure.

Additional comments

Included in Basic reporting.

Cite this review as

·

Basic reporting

This work has proposed an intelligent zero trust framework based on SDN. The authors have done evaluation with datasets. However, there are few issues which need to be addressed:

a) Introduction, related work should be rephrased. It is difficult to understand.
b) Figures and tables should be provided in the section where they have been mentioned.
c) In the related work authors can describe how this work is going to be different than what they have described in the background studies.

Experimental design

1) In the experimental design authors have used iperf, they can use some other tools such as HPING, Botnet simulator to simulate the attack.
2) They can provide explanation why they have used RYU controller instead of some other SDN controller.
3) The authors can provide a separate section describing the experimental settings with clear explanation.

Validity of the findings

The authors have provided clear figure and explanation for the results. However, they should address few issues:
1) The authors can provide some comparison with previous work on zero trust architecture using SDN.
2) Why this framework is better than others.

Reviewer 4 ·

Basic reporting

This paper proposes an intelligent zero-trust security framework (IZTSDN) for Software-Defined Networking (SDN) by integrating deep learning and zero-trust architecture. The framework aims to protect every resource and network connection in the network by analyzing users' network behavior in real-time and restricting malicious users from accessing network resources. The proposed IZTSDN framework consists of five modules: data collection, trust assessment engine, intelligent behavior analysis engine, intelligent controller, and communication execution. It's clear, understandable, self-contained, and well-structured.

Experimental design

The experiment is well-designed. The experimental results demonstrated that the CALSeq2Seq traffic anomaly detection model performed well in terms of accuracy, precision, recall, and F1-measure. It provided continuous and accurate detection of user behavior monitoring and identification, contributing to the security of SDN networks.

Validity of the findings

no comment

Cite this review as

---

## Round 0.2 · accepted · Accept

Through the integration of deep learning and zero-trust architecture, we offer an intelligent zero-trust security framework for software-defined networks (IZTSDN) in this study. This framework uses zero-trust architecture to secure all network connections and resources.

This paper has a good contribution to the research community and it is worth publishing in PeerJ Computer Science. The paper in a very good level.

Reviewer 2 ·

Basic reporting

The suggested changes are incorporated.

Experimental design

The authors must change the val ---> validation and train ----> training in Figures 11 and 12.

Validity of the findings

The suggested changes are incorporated.

Additional comments

No comments

Cite this review as